# Association between Selected Dietary Habits and Lipid Profiles of Patients with Cardiovascular Disease

**DOI:** 10.3390/ijerph17207605

**Published:** 2020-10-19

**Authors:** Jana Kopčeková, Mária Holovičová, Martina Gažarová, Jana Mrázová, Marta Habánová, Lucia Mečiarová, Monika Bronkowska

**Affiliations:** 1Department of Human Nutrition, Faculty of Agrobiology and Food Resources, Slovak University of Agriculture in Nitra, 949 01 Nitra, Slovakia; jana.kopcekova@uniag.sk (J.K.); mariaholovicova23@gmail.com (M.H.); martina.gazarova@uniag.sk (M.G.); jana.mrazova@uniag.sk (J.M.); marta.habanova@uniag.sk (M.H.); lucia.meciarova@leanenterprise.sk (L.M.); 2Department of Human Nutrition, Faculty of Biotechnology and Food Science, Wroclaw University of Environmental and Life Sciences, 51-630 Wroclaw, Poland

**Keywords:** cardiovascular disease, lipid profile, food frequency, dietary habits

## Abstract

This study evaluated the associations between selected dietary habits and lipid profiles in a group of 800 randomly selected patients hospitalized in the Nitra Cardio Center, Slovakia. Patients were aged 20–101 years (only men, the average age was 61.13 ± 10.47 years). The data necessary for the detection of dietary habits were obtained by a questionnaire method in closed-ended format. Data collection was carried out simultaneously with the somatometric and biochemical examinations of the respondents ensured by the Nitra Cardio Center. The following parameters were evaluated: total cholesterol (T-C), low-density lipoprotein cholesterol (LDL-C), high-density lipoprotein cholesterol (HDL-C), triglycerides, and blood glucose. Statistical comparisons between groups were performed using one-way analysis of variance (one-way ANOVA), followed by Tukey’s post hoc test. We detected significant differences (*p* < 0.05) in the influence of the number of daily meals on T-C and LDL-C, which were higher in men who consumed 1–2 meals compared with 3–4 or 5–6 meals. In the consumption of meat, eggs, and fish, there was no significant effect on the biochemical parameters of blood (*p* > 0.05). We recorded a significant effect (*p* < 0.001) on T-C and LDL-C levels between low-fat and whole-fat milk consumption. Except for the impact of fruit consumption on the HDL-C level (*p* < 0.001), the different frequencies of fruit consumption showed non-significant changes for the lipid profile levels. We detected a significant effect (*p* = 0.017) of the consumption of vegetables 1–2 times/week on LDL-C in favor of daily consumption. Our results support that monitoring the lipid profile is an important determinant in the prevention and treatment of cardiovascular disease. The conducted research emphasizes the importance of diet dependence on the improvement of the quality of treatment and nutrition of people with this type of disease.

## 1. Introduction

Currently, a substantial increase in cardiovascular disease (CVD) is noticeable all over the world; it is deemed to be the main cause of death in both developed and developing countries [1,2]. The proportion of deaths in Europe from CVD amongst men is 43% and 55% amongst women. It is estimated that 30% of people under the age of 65 on this continent die of CVD [3]. The main cause of CVD (which is the reason for high mortality in the population) is atherosclerosis. The causes and risk factors of atherosclerosis are still not fully known. In spite of this, some risk determinants have been identified that may increase the risk of developing atherosclerosis. Mainly, they include: high-serum total cholesterol (T-C), low-density lipoprotein cholesterol (LDL-C), and high-density lipoprotein cholesterol (HDL-C), as well as hypertension, smoking, diabetes, obesity, and sedentary lifestyle [4].

Numerous clinical and epidemiological experimental studies have shown that nutrition plays an important role in the prevention and secondary prevention of atherosclerosis [4]. It is assumed that as much as 80% of cardiovascular incidents could probably be avoided by implementing a proper lifestyle (weight control, avoiding alcohol and smoking, exercising, and balanced nutrition). It is also necessary to pay attention to adequate clinical care and biological risk factors [3]. The World Health Organization emphasizes that improper nutrition, a sedentary lifestyle, alcohol abuse, and smoking are the main modifiable risk factors for chronic diseases such as CVD, cancer, respiratory disorders, and metabolic diseases [5].

The first evidence of the association between diet/physical activity and well-being was found in the 1960s following the first results of a study named the “Seven Countries’ Study” [6]. The first results from this study were surprising because they distinctly showed that the countries of the Mediterranean basin (Italy and Greece) had the lowest incidence of death from CVD and cancer compared to all other countries [6]. In the last two decades, considerable knowledge has been gained from intervention studies about food and nutrients that favorably or adversely affect the cardiovascular system [7,8].

While foods from animal sources contain important nutrients that may not be readily available from plant sources, high consumption of red meat is associated with a greater risk of morbidity and mortality from CVD. Such diets may contain high levels of total and saturated fats in addition to cholesterol. Numerous studies have demonstrated the benefits of diets high in fruit, vegetables, whole grains, and fish, as well as low in red meat, high-fat dairy products, and trans- and saturated fats. On the other hand, most recent meta-analyses of randomized trials and observational studies found no beneficial effects of reducing saturated fatty acid (SFA) intake on CVD and total mortality, and instead found protective effects against strokes [9].

High concentrations of T-C, LDL cholesterol, and triacylglycerols indicate that the individual is at risk for the onset of CVD [10]. Increased concentration of LDL cholesterol in the blood serum is a recognized risk factor for the development of heart disease. However, the relationship between the dietary cholesterol content and the serum LDL-C concentration is still unknown [11].

This study aimed to evaluate the associations between selected dietary habits and lipid profiles in a group of randomly selected patients hospitalized in the Nitra Cardio Center, Slovakia.

The research was approved by the Ethics Committee of Kardiocentrum Nitra and the Ethics Committee of the Specialized Hospital, St. Zoerardus Zobor, protocol number 10.6.2014.

## 2. Materials and Methods

We evaluated the relationship between selected dietary habits and lipid profiles in a group of 800 randomly selected men hospitalized in the Nitra Cardio Center, Slovakia. The study was approved by the Nitra Cardio Center, and all participants signed the written consent before participating in the study. Patients were aged 20–101 years (the average age was 61.13 ± 10.47 years). Patients were selected using the method of random selection. Respondents included in the study had either overcome myocardial infarction or were diagnosed with angina pectoris and hospitalized after undergoing catheterization. Data were collected during the years 2015–2018. 

The data necessary for the detection of dietary habits were obtained by a questionnaire method in closed-ended format. The questionnaire was applied individually by a single interviewer. The questionnaire contained two parts. The first part included questions concerning the socio-demographic situation of the subjects, physical activity, use of tobacco, and medical history. The second part of the questionnaire concerned the analysis of selected dietary habits, including the number of the consumed meals, their regularity, snacking between meals, types of snacks consumed, and the eating frequency of selected groups of food products. The questionnaire was anonymous; its completion was voluntary, with only one response to be circled for each question. The questionnaire was compiled by the Department of Human Nutrition (Slovak University of Agriculture in Nitra) and approved by the Nitra Cardio Center.

Data collection was carried out simultaneously with the somatometric and biochemical examinations of the respondents ensured by the Nitra Cardio Center. The following parameters, considered to be major risk factors for CVD, were evaluated: T-C, LDL-C, HDL-C, triglycerides (TG), and blood glucose. Venous blood was collected in the morning after 8 h of fasting in a standard manner, using 2.5 mL EDTA (ethylenediaminetetraacetic acid) solution in a 7.5 mL serum gel tube. Following the separation of blood serum, the lipid profile was measured using the Automatic Biochemical Analyzer BioMajesty^®^ JCA-BM6010/C (DiaSys Diagnostic System GmbH, Holzheim, Germany). The anthropometric parameters of body weight (kg) and height (cm) were measured on outpatient electronic medical scales (Tanita WB-3000, Tanita Co., Tokyo, Japan). The body mass index (BMI) was calculated by dividing the body weight in kilograms by the square of the height in meters.

The data were checked for normality. Assumptions for the normal distribution of data were tested using the Shapiro–Wilk test. Parametric statistics were used to evaluate differences in parameters that were characterized by a normal distribution (Food Frequency Questionnaire - FFQ). Statistical comparisons between groups were performed using one-way analysis of variance (ANOVA), followed by Tukey’s post hoc test. For parameters inconsistent with the normal distribution, they were expressed as the median (Me) and the lower and upper quartiles (Q1, Q3). Here, nonparametric statistics and Spearman’s rank order correlations were used to compare the differences. Two-sample non-parametric comparisons were performed using the Mann–Whitney U test. Significance was assumed when *p* < 0.05. Statistica Cz 10 (TIBCO Software, Inc., Palo Alto, CA, USA) and MS Excel 2007 (Microsoft Corporation, Redmond, WA, USA) were used.

## 3. Results

From the obtained individual values, we calculated the basic statistical characteristics. These characteristics of the study participants are shown in Table 1. The demographic characteristics of study participants are shown in Table 2.

In the prevention of obesity and lipid metabolism disorders, attention is paid to the regular consumption of meals, with appropriate time intervals between them. This statement is supported by the fact that people who consume low-quantity food doses per day had increased levels of T-C and a higher prevalence of obesity and CVD [12], which was also confirmed by our results. It turned out that most of the respondents (69.25%) consumed 3–4 meals a day (Table 3). Only 25.3% of participants reported 5–6 meals a day. We detected significant differences (*p* ˂ 0.05) between the influence of the number of daily meals on T-C and LDL-C, which were higher in men who consumed 1–2 meals compared with 3–4 or 5–6 meals (Table 3). Previous studies have shown that a higher daily eating frequency is associated with an increased intake of energy, which comes mainly from carbohydrates and not from fats or proteins. Such studies also found a positive relationship between a higher daily eating frequency and the consumption of nutrients such as dietary fiber, vitamin C, folic acid, calcium, magnesium, and iron [13].

Although consumption of red meat is valuable as a source of protein, iron, vitamin B_12_, and other B vitamins in the human diet [14], its excessive consumption, as seen in many Western nations, may be associated with obesity, CVD, some types of oncological diseases, diabetes, and other non-communicable diseases [15,16,17]. The link between meat consumption and the incidence of chronic diseases and mortality has been evaluated in hundreds of observational epidemiological studies over the last few decades. Despite these substantial data, it is not yet clear whether the increased intake of certain groups of meat (e.g., total, unprocessed, or processed red meat) or certain species (e.g., beef or pork) contribute independently to the risk of the disease [18].

We found that pork is the most popular; 72.7% of respondents reported consuming it 1–2 times/week (Table 4). Escriba-Perez et al. [19] monitored the consumption of meat in 800 respondents aged 25–75 years. The most common was chicken, which was consumed by up to 90.87% of respondents at least once a week. The second most frequently consumed was beef (63.62%), and third was pork (52.62%). Escriba-Perez et al. [19] also stated that red meat consumption frequency also differed between the sexes. Beef was consumed more frequently by males, while females consumed more chicken than red meat. The results of the effect of the frequency of red meat consumption on the lipid profiles of participants in the current study are shown in Table 4. The highest T-C and LDL-C values were seen in men who consumed red meat 3–4 times/week. For the consumption of pork and beef, there was no significant effect on the blood biochemical parameters (*p* > 0.05). Only pork consumption had a significant effect on TG (*p* = 0.046). Kontogianni et al. [20] found that a high intake of red meat (more than eight servings/month) was associated with an increased risk of acute coronary syndromes, but low intake (less than four servings/month) showed no association.

Our results show that most respondents (60.5%) consume 1–5 eggs weekly (Table 5). This consumption rate is associated with non-significantly lower T-C and LDL-C (*p* > 0.05). The significant effect (*p* < 0.001) was observed on the TG levels, which were highest at the consumption of >10 eggs per week. Consumption of more eggs was non-significantly associated with higher T-C and LDL-C and lower HDL-C values. Baumgartner et al. [21] found that in 57 healthy volunteers (32 females and 25 men) aged 18–65 years, the consumption of one egg/day increased T-C by 0.63 mmol/L and LDL-C by 0.59 mmol/L relative to the control group. On the contrary, in a cohort study with 7216 participants, Díez-Espino et al. [22] found that consumption of 2–4 eggs/week was not associated with high cardiovascular risk. From a meta-analysis based on randomized controlled trials, with intervention periods ranging between 3 and 20 weeks and changes in cholesterol intakes between 137 and 897 mg/day, it was estimated that each 100 mg increase in dietary cholesterol intake increased the serum LDL-C concentration by 0.050 mmol/L [23].

In our group of respondents, the consumption of fish (sea or freshwater, 1–2 times/week) was recorded in only 35% and 43.1% of men, respectively (Table 6). Alarmingly, up to 24% of the patients consumed sea fish only 1–2 times/month. The different frequencies of fish consumption showed non-significant changes regarding the lipid profile. A meta-analysis of 17 cohort studies showed that both low (one serving/week) and moderate (2–4 servings/week) fish consumption were inversely associated with coronary heart disease (CHD) mortality, while no additional benefits were seen at higher intakes [24].

The daily consumption of milk and milk products was recorded in 45.2% of our patients (Table 7). The results indicated that T-C, LDL-C, and TG values were highest in patients who consumed medium-fat milk. We recorded a significant effect of milk type on the T-C and LDL-C levels.

Among the 800 participants in our study, 64.7% ate fruit and 54.7% ate vegetables as part of their daily diet (Table 8). Respondents who consumed fruits and vegetables daily had higher T-C and LDL-C values. Except for the impact of fruit consumption on HDL-C levels (*p* < 0.001), the different frequencies of fruit consumption showed non-significant changes in the lipid levels. We detected a significant effect (*p* = 0.017) of the consumption of vegetables 1–2 times/week on LDL-C in favor of daily consumption. 

The study group also analyzed the correlation of Spearman’s rank order (r) with division into age groups, BMI value, and smoking status. Significance was assumed when *p* < 0.05.

Table 9 presents the Kruskal–Wallis rank ANOVA by age group. No statistically significant correlation was found in this analysis.

Next, an ANOVA of the Kruskal–Wallis ranks was performed with the division of the BMI index range (Table 10). Significance was assumed when *p* < 0.05. No statistically significant correlation was found in this analysis.

Next, an ANOVA of the Kruskal–Wallis ranks was performed with the sample divided into smokers and non-smokers (Table 11). Significance was assumed when *p* < 0.05. No statistically significant correlation was found in this analysis.

In the next stage, the Mann–Whitney U test was carried out, taking into account the status of smoking cigarettes. Significance was assumed when *p* < 0.05. There were statistically significant differences in glucose concentration between smokers and non-smokers (Table 11).

Table 12 shows the results of the significance of Spearman’s rank correlation (r > 0.5) by group. Significance was assumed when *p* < 0.05. Spearman’s rank correlations were found to be statistically significant (r > 0.5). 

It was found that there were statistically significant correlations between the selected biochemical indices. With the increase in glucose, the concentration of HDL-cholesterol fraction drops statistically significantly (r = −0.21). As the concentration of triglycerides increases, the concentration of the HDL cholesterol ratio decreases (r = −0.26).

Due to the fact that in the age/BMI/smoker groups, no equal variances were found, the research hypothesis did not allow for a two-factor analysis.

## 4. Discussion

The European Prospective Investigation into Cancer and Nutrition (EPIC) study included nearly 500,000 participants across 10 European countries and more than 5000 cardiovascular events. It confirmed that the consumption of processed meat is strongly associated with CVD risk and that the consumption of unprocessed red meat has little to no association [25].

In a meta-analysis of 20 studies (17 prospective cohorts and three case–control studies) that included 1,218,380 individuals, Micha et al. [26] concluded that intake of processed, but not red, meat was associated with an increased incidence of CHD. The authors further speculated that the higher sodium and nitrate content of processed meat might contribute to its impact on CVD.

Since the discovery that eggs, and especially egg yolks, contain a notable quantity of cholesterol, numerous studies have tried to investigate the effect of egg consumption on the risk of CVD. Following the results of these studies, it is now widely known that dietary cholesterol intake does not noticeably affect fasting plasma lipid profile, and some studies showed that eggs might even have anti-atherogenic properties [27]. Although eggs are a major source of dietary cholesterol, they are also an inexpensive source of unsaturated fat, high-quality protein, folate, and other vitamins and minerals [28].

Many studies suggest that increasing fish consumption is recommended for the intake of omega-3 fatty acids and also to confer benefits for CVD risk reduction [29]. In a prospective analysis on 20,969 subjects free from CVD at baseline, Bonaccio et al. [30] detected that fish was consumed by almost all subjects in the cohort (99.5%). Beyond promoting a healthy dietary lifestyle overall, dietary guidelines recommend eating fish at least twice (two servings) a week [30] because of the well-documented inverse association with risk of developing CVD (supported by several meta-analyses). Mean total fish intake in the study sample was 44.6 g/day, and mean frequency of fish intake was 2.4 times/week.

The results of the observational study, regardless of the level of milk fat, have not found a link between milk product intake and increased risk of CHD, stroke, or other heart and vessel diseases. Short-term intervention studies have shown that whole milk and butter increase not only LDL-C, but also HDL-C, and, therefore, may not affect plasma levels (or could reduce T-C to HDL-C) [31]. Milk and dairy products are a source of energy and high-quality protein, and they also contain important trace elements in our diet [32]. Consumption of milk and its products is one of the key elements of a rational, balanced diet. In the case of mammals, milk is the first food that provides energy and all the necessary nutrients to ensure proper growth and development. This highlights the very important role of milk and its products as a source of calcium, which plays a key role in the formation of bone mass. However, there is some controversy about the consumption of dairy and dairy products in adulthood, especially concerning non-human milk. Despite these controversies, epidemiological studies support the possible role of its consumption in preventing many chronic diseases, such as CVD, some forms of cancer, obesity, and diabetes [33]. The association of milk or dairy product consumption with CVD is still controversial [34,35].

Milk and dairy products have been shown to have a beneficial effect on lowering blood pressure and increasing HDL-C, which is associated with a reduced risk of CVD [32]. Most dietary guidelines recommend 2–3 servings of dairy products/day [36,37]. More recently, Drouin-Chartier et al. [38] conducted a randomized control trial involving 76 patients with mild to moderate hypertension. It was found that consuming a total of three servings per day of milk (low-fat), yoghurt (low-fat), and cheese (regular-fat) significantly reduced the mean daytime systolic blood pressure (2 mmHg; *p* = 0.05) in men but not in women when compared with a dairy-free control diet. In terms of cardiovascular health, however, the fat content of these products is important [31].

Fruits and vegetables are highly nutritious products, containing large amounts of dietary fiber, vitamins, and minerals, and are also characterized by a high antioxidant potential. It is believed that higher consumption of these product groups reduces the risk of developing chronic diseases, including diseases of the cardiovascular system. Therefore, it is recommended to include five portions of fruits and vegetables in daily food rations [39]. From randomized controlled studies, fruits and vegetables were shown to have a favorable effect on several risk factors, including: blood pressure, lipid concentration, insulin resistance, inflammatory biomarker concentrations, endothelial function, and weight control [40,41].

He et al. [41] reported that the average fruit and vegetable intake in most developed countries is three portions/day, which is below the recommended amount of at least five portions/day. Pérez [42] states that women consume fruits and vegetables more often than men, and men with diagnosed cancer, heart disease, high blood pressure, or diabetes mellitus tend to consume these foods more often than healthy men.

Although this study provided new insight into the relationship between selected eating habits and the lipid profiles of patients with cardiovascular disease, it has some limitations. First, the study was conducted only on the population of one country, so it provides detailed information only about that particular population, and should be reproduced in other countries. Moreover, the study only looked at men, while, in fact, the relationship between eating habits is, of course, also relevant among women. In subsequent studies, it is worth conducting similar studies among people of both sexes. Additionally, comparisons of the results obtained by gender would probably be very valuable. Another limitation of the publication may also be unequal age groups, unequal groups, and a very large discrepancy within the group when it comes to the respondents. Finally, the study only assessed the declared behaviors, which were not verified by direct observational research techniques.

## 5. Conclusions

In our study, we observed the associations between selected dietary habits and lipid profiles in a group of randomly selected patients with CVD. Our results showed a positive relationship between a higher daily eating frequency and lower levels of T-C and LDL-C. We have not confirmed the effect of more frequent consumption of red meat on higher T-C and LDL-C levels. Consumption of more than 10 eggs was associated with higher T-C, LDL-C, and lower HDL-C levels. We recorded a significant difference between the effect of low-fat and whole-fat milk consumption on the T-C and LDL-C levels. Except for the impact of fruit consumption on HDL-C level, the different frequencies of consumption of fruit showed non-significant changes regarding the profile of lipid levels. We found a positive effect of daily consumption of vegetables on LDL-C, compared with consumption 1–2 times/week. Our results support the hypothesis that diet is one of the most important determinants in CVD and its associated risk factors.

## Figures and Tables

**Table 1 ijerph-17-07605-t001:** Characteristics of study participants (*n* = 800).

Characteristic	Me	Q1	Q3
Age (years)	61.00	56.00	68.00
BMI (kg/m^2^)	29.05	26.73	31.78
TC (mmol/L)	4.65	3.80	5.55
LDL-C (mmol/L)	2.85	2.19	3.52
HDL-C (mmol/L)	1.11	0.92	4.01
TG (mmol/L)	1.51	1.07	2.21
Glucose (mmol/L)	5.94	5.30	7.28

Abbreviations: Me: median; Q1: lower quartile; Q3: upper quartile; BMI: body mass index; T-C: total cholesterol; LDL-C: low-density lipoprotein cholesterol; HDL-C: high-density lipoprotein cholesterol; TG: triglycerides.

**Table 2 ijerph-17-07605-t002:** Demographic characteristics of study participants (*n* = 800).

Characteristics	*n*	%	Characteristics	*n*	%
Social status			Family status		
employed	273	34.2	married	582	72.7
unemployed	127	15.7	divorced	111	13.9
retired	400	50.1	widowed	107	13.4
Age categories (years)			Education		
˂40	30	3.7	basic	98	12.2
40–49	65	8.1	apprenticeship	234	29.4
50–59	229	28.6	secondary	302	37.7
60–69	315	39.4	higher	166	20.7
70–79	147	18.4	Physical activity		
≥80	14	1.8	15–30 min/day	285	35.6
BMI (kg/m^2^)			30–60 min/day	161	20.1
˂18.5	3	0.3	>60 min/day	354	44.3
18.5–25	93	11.6	Smoker		
25–30	402	50.2	Yes	251	31.4
≥30	302	37.9	No	549	68.6

**Table 3 ijerph-17-07605-t003:** Effect of the number of daily meals on lipid profile (mmol/L).

Number of Daily Meals	*n* (%)	T-C (Average ± SD)	LDL-C (Average ± SD)	HDL-C (Average ± SD)	TG (Average ± SD)
1–2	76 (9.5)	4.92 ± 1.13	3.14 ± 0.84	1.19 ± 0.42	1.59 ± 0.80
3–4	522 (65.2)	4.73 ± 1.20	2.98 ± 1.02	1.16 ± 0.40	1.77 ± 1.03
5–6	202 (25.3)	4.34 ± 1.12	2.65 ± 0.90	1.11 ± 0.35	1.64 ± 0.94
*p*-value		0.031 ^a^; 0.033 ^b^	0.032 ^a^; 0.031 ^b^	>0.05	>0.05

Abbreviations: T-C: total cholesterol; LDL-C: LDL cholesterol; HDL-C: HDL cholesterol; TG: triglycerides. ^a^ Significant difference between 1–2 and 5–6 daily meals. ^b^ Significant difference between 3–4 and 5–6 daily meals.

**Table 4 ijerph-17-07605-t004:** Effect of the frequency of consumption of red meat on lipid profile (mmol/L).

Frequency of Meat Consumption	*n* (%)	T-C (Average ± SD)	LDL-C (Average ± SD)	HDL-C (Average ± SD)	TG (Average ± SD)
*Pork*					
1–2 times/week	582 (72.7)	4.57 ± 1.11	2.82 ± 0.93	1.19 ± 0.46	1.68 ± 0.82
3–4 times/week	94 (11.7)	4.78 ± 1.07	2.87 ± 0.88	1.14 ± 0.33	2.02 ± 0.98
1–2 times/month	112 (14.0)	4.77 ± 1.33	2.92 ± 0.98	1.17 ± 0.28	1.94 ± 1.08
no consumption	12 (1.6)	4.58 ± 1.09	2.78 ± 0.87	1.18 ± 0.09	1.96 ± 0.37
*p*-value		>0.05	>0.05	>0.05	0.046 ^a^, 0.039 ^b^
*Beef*					
1–2 times/week	370 (46.2)	4.62 ± 1.14	2.84 ± 0.91	1.17 ± 0.38	1.68 ± 0.77
3–4 times/week	44 (5.5)	4.52 ± 0.78	2.70 ± 0.38	1.07 ± 0.29	1.99 ± 1.23
1–2 times/month	374 (46.7)	4.51 ± 1.16	2.79 ± 1.01	1.21 ± 0.49	1.77 ± 0.88
no consumption	12 (1.6)	4.53 ± 0.18	3.08 ± 0.78	1.17 ± 0.06	1.87 ± 0.13
*p*-value		>0.05	>0.05	>0.05	>0.05

Abbreviations: T-C: total cholesterol; LDL-C: LDL cholesterol; HDL-C: HDL cholesterol; TG: triglycerides. ^a^ Significant difference between 1–2 times/week and 3–4 times/week. ^b^ Significant difference between 1–2 times/week and 1–2 times/month.

**Table 5 ijerph-17-07605-t005:** Effect of consumption of eggs on lipid profile (mmol/L).

Number of Eggs/Week	*n* (%)	T-C (Average ± SD)	LDL-C (Average ± SD)	HDL-C (Average ± SD)	TG (Average ± SD)
no consumption	162 (20.2)	4.06 ± 1.02	2.43 ± 0.94	1.16 ± 0.31	1.45 ± 0.64
1–5	484 (60.5)	4.59 ± 1.10	2.77 ± 0.88	1.17 ± 0.34	1.74 ± 0.88
6–10	134 (16.7)	4.81 ± 1.16	2.92 ± 0.78	1.19 ± 0.44	2.01 ± 1.00
>10	20 (2.6)	5.30 ± 0.57	3.34 ± 0.28	0.97 ± 1.11	3.81 ± 1.06
*p*-value		>0.05	>0.05	>0.05	<0.001 ^a,b^

Abbreviations: T-C: total cholesterol; LDL-C: LDL cholesterol; HDL-C: HDL cholesterol; TG: triglycerides. ^a^ Significant difference between 1–5 eggs and more than 10 eggs/week. ^b^ Significant difference between 6–10 eggs and more than 10 eggs/week.

**Table 6 ijerph-17-07605-t006:** Effect of the frequency of consumption of fish on lipid profile.

Frequency of Consumption	*n* (%)	T-C (Average ± SD)	LDL-C (Average ± SD)	HDL-C (Average ± SD)	TG (Average ± SD)
*Sea fish*					
1–2 times/week	280 (35.0)	4.64 ± 1.23	2.80 ± 0.97	1.23 ± 0.53	1.70 ± 0.82
1–2 times/month	192 (24.0)	4.48 ± 1.20	2.89 ± 0.97	1.20 ± 0.30	1.54 ± 0.81
sometimes	136 (17.0)	4.49 ± 0.97	2.69 ± 0.79	1.17 ± 0.39	1.78 ± 0.90
no consumption	192 (24.0)	4.65 ± 1.14	2.96 ± 0.98	1.15 ± 0.26	1.98 ± 0.96
*p*-value		>0.05	>0.05	>0.05	0.001 ^a^, 0.022 ^b^
*Freshwater fish*					
1–2 times/week	345 (43.1)	4.70 ± 1.26	2.97 ± 1.03	1.15 ± 0.40	1.70 ± 1.08
1–2 times/month	293 (36.6)	4.73 ± 1.16	2.97 ±1.00	1.15 ± 0.39	1.67 ± 0.91
sometimes	105 (13.1)	4.53 ± 1.10	2.76 ± 0.98	1.15 ± 0.36	1.94 ± 0.98
no consumption	57 (7.2)	4.21 ± 0.99	2.54 ± 0.69	1.11 ± 0.31	1.79 ± 0.83
*p*-value		>0.05	>0.05	>0.05	>0.05

Abbreviations: T-C: total cholesterol; LDL-C: LDL cholesterol; HDL-C: HDL cholesterol; TG: triglycerides. ^a^ Significant difference between 1–2 times/month and no consumption. ^b^ Significant difference between 1–2 times/week and no consumption.

**Table 7 ijerph-17-07605-t007:** Effect of milk type and frequency of consumption of milk and dairy products on lipid profile.

Parameter	*n* (%)	T-C (Average ± SD)	LDL-C (Average ± SD)	HDL-C (Average ± SD)	TG (Average ± SD)
*Type of Milk*					
low-fat	86 (10.7)	4.41 ± 1.26	3.32 ± 1.03	1.20 ± 0.45	1.82 ± 1.06
medium-fat	334 (41.7)	4.52 ± 1.15	2.77 ± 0.97	1.13 ± 0.39	1.59 ± 0.83
whole-fat	173 (21.6)	4.42 ± 1.16	2.64 ± 0.98	1.16 ± 0.42	1.67 ± 0.84
no consumption	207 (26.0)	4.60 ± 1.17	2.87 ± 0.96	1.19 ± 0.37	1.83 ± 1.06
*p*-value		<0.001 ^a,b^, 0.006 ^c^	0.002 ^a^, <0.001 ^b^, 0.015 ^c^	>0.05	>0.05
*Frequency of Dairy Product Consumption*					
daily	362 (45.2)	4.90 ± 1.22	2.81 ± 1.01	1.49 ± 0.57	1.70 ± 1.37
3–4 times/week	166 (20.7)	4.52 ± 0.99	2.48 ± 0.97	1.38 ± 0.30	1.44 ± 0.65
1–2 times/week	248 (31.0)	4.82 ± 1.52	2.82 ± 1.12	1.45 ± 0.42	1.59 ± 0.90
no consumption	24 (3.1)	5.03 ± 0.93	3.01 ± 0.65	1.21 ± 0.44	2.40 ± 1.54
*p*-value		>0.05	>0.05	>0.05	>0.05

Abbreviations: T-C: total cholesterol; LDL-C: LDL cholesterol; HDL-C: HDL cholesterol; TG: triglycerides. ^a^ Significant difference between medium- fat and low-fat milk. ^b^ Significant difference between whole-fat and low-fat milk. ^c^ Significant difference between no consumption and low-fat milk.

**Table 8 ijerph-17-07605-t008:** Effect of the frequency of consumption of fruit and vegetables on lipid profile.

Type of Food	*n* (%)	T-C (Average ± SD)	LDL-C (Average ± SD)	HDL-C (Average ± SD)	TG (Average ± SD)
*Fruit*					
daily	518 (64.7)	4.60 ± 1.17	2.89 ± 0.98	1.13 ± 0.37	1.63 ± 0.83
3–4 times/week	146 (18.2)	4.55 ± 1.22	2.80 ± 0.94	1.14 ± 0.37	1.71 ± 0.82
1–2 times/week	120 (15.0)	4.64 ± 1.49	2.64 ± 1.20	1.43 ± 0.43	1.38 ± 0.95
no consumption	16 (2.1)	4.20 ± 0.61	2.57 ± 0.62	1.20 ± 0.27	1.53 ± 0.62
*p*-value		>0.05	>0.05	<0.001 ^a,b^	>0.05
*Vegetable*					
daily	438 (54.7)	4.79 ± 1.12	2.72 ± 0.89	1.45 ± 0.51	1.76 ± 1.11
3–4 times/week	170 (21.3)	5.02 ± 1.32	3.02 ± 1.13	1.43 ± 0.37	1.40 ± 0.63
1–2 times/week	137 (17.1)	5.14 ± 1.37	3.09 ± 1.15	1.49 ± 0.37	1.46 ± 1.22
no consumption	55 (6.9)	4.58 ± 1.30	2.70 ± 0.99	1.20 ± 0.32	1.78 ± 0.59
*p*-value		>0.05	0.017 ^a^	>0.05	>0.05

Abbreviations: T-C: total cholesterol; LDL-C: LDL cholesterol; HDL-C: HDL cholesterol; TG: triglycerides. ^a^ Significant difference between daily and 1–2 times/week. ^b^ Significant difference between 3–4 times/week and 1–2 times/week.

**Table 9 ijerph-17-07605-t009:** Kruskal–Wallis rank analysis of variance (ANOVA) by age group.

Age	*n*	Glucose Me (Q1; Q3)	T-C Me (Q1; Q3)	LDL-C Me (Q1; Q3)	HDL-C Me (Q1; Q3)	TG Me (Q1; Q3)	BMI Me (Q1; Q3)
<30	8	5.34	3.13	2.05	0.79	1.31	26.77
		(4.77; 7.15)	(3.00; 5.05)	(1.95; 3.24)	(0.72; 1.05)	(1.01; 2.58)	(17.90; 28.06)
31–55	105	5.95	5.95	5.95	5.95	5.95	5.95
		(5.31; 7.20)	(5.31; 7.20)	(5.31; 7.20)	(5.31; 7.20)	(5.31; 7.20)	(5.31; 7.20)
56–75	616	5.93	4.60	2.84	1.10	1.51	29.06
		(5.31; 7.27)	(3.79; 5.54)	(2.19; 3.48)	(0.91; 1.34)	(1.10; 2.21)	(26.83; 31.38)
>75	71	6.00	4.83	3.11	1.10	1.39	29.05
		(5.18; 7.40)	(3.92; 5.99)	(2.15; 3.72)	(0.91; 1.34)	(0.88; 2.18)	(26.85; 30.45)
*p*-value		>0.05	>0.05	>0.05	>0.05	>0.05	>0.05

Abbreviations: Me: median; Q1: lower quartile; Q3: upper quartile; BMI: body mass index; T-C: total cholesterol; LDL-C: LDL cholesterol; HDL-C: HDL cholesterol; TG: triglycerides.

**Table 10 ijerph-17-07605-t010:** Kruskal–Wallis rank ANOVA analysis by BMI group.

BMI	*n*	Glucose Me (Q1; Q3)	T-C Me (Q1; Q3)	LDL-C Me (Q1; Q3)	HDL-C Me (Q1; Q3)	TG Me (Q1; Q3)
<18.49	3	4.77 (4.77; 5.30)	3.13 (3.13; 6.93)	2.05 (2.05; 3.85)	0.72 (0.72; 1.13)	1.24 (1.24; 3.48)
18.5–24.99	93	6.40 (5.22; 7.20)	5.95 (5.31; 7.20)	5.95 (5.31; 7.20)	5.95 (5.31; 7.20)	5.95 (5.31; 7.20)
25–29.99	401	6.00 (5.31; 7.27)	4.60 (3.79; 5.54)	2.84 (2.19; 3.48)	1.10 (0.91; 1.34)	1.51 (1.10; 2.21)
>30	303	5.90 (5.31; 7.20)	4.72 (3.83; 5.55)	2.83 (2.20; 3.73)	1.10 (0.89; 1.36)	1.66 (1.16; 2.35)
*p*-value		>0.05	>0.05	>0.05	>0.05	>0.05

Abbreviations: Me: median; Q1: lower quartile; Q3: upper quartile; BMI: body mass index; T-C: total cholesterol; LDL-C: LDL cholesterol; HDL-C: HDL cholesterol; TG: triglycerides.

**Table 11 ijerph-17-07605-t011:** Kruskal–Wallis rank ANOVA analysis by smoker group.

Smoker	*n*	Glucose Me (Q1; Q3)	T-C Me (Q1; Q3)	LDL-C Me (Q1; Q3)	HDL-C Me (Q1; Q3)	TG Me (Q1; Q3)
no	532	5.88 (5.38; 7.20) *	4.71 (3.83; 5.57)	2.87 (2.21; 3.54)	1.11 (0.94; 1.36)	1.52 (1.05; 2.27)
yes	268	6.14 (5.22; 7.40) *	4.50 (3.79; 5.46)	2.83 (2.12; 3.45)	1.09 (0.91; 1.36)	1.46 (1.13; 2.04)
*p*-value		<0.03 *	>0.05	>0.05	>0.05	>0.05

Abbreviations: Me: median; Q1: lower quartile; Q3: upper quartile; BMI: body mass index; T-C: total cholesterol; LDL-C: LDL cholesterol; HDL-C: HDL cholesterol; TG: triglycerides; * Kruskal–Wallis ranks.

**Table 12 ijerph-17-07605-t012:** Correlations by Spearman’s rank.

	Glucose	T-C	LDL-C	HDL-C	TG
**Glucose**	-	−0.047	−0.032	**−0.21 ***	0.067
**T-C**	−0.047	-	**0.87 ***	**0.27 ***	**0.26 ***
**LDL-C**	−0.032	**0.87 ***	-	**0.10 ***	**0.16 ***
**HDL-C**	**−0.21 ***	**0.27 ***	**0.10 ***	-	**−0.26 ***
**TG**	0.067	**0.26 ***	**0.16 ***	**−0.26 ***	-

Spearman’s rank correlations. Significance was assumed when *p* < 0.05 *; Bold, the correlations statistically significant in the analysis of Spearman's rank correlation at the significance level *p* > 0.5.

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
