# Peer review of "Association between Selected Dietary Habits and Lipid Profiles of Patients with Cardiovascular Disease"

_ijerph, 2020, doi:10.3390/ijerph17207605_

Round 1

Reviewer 1 Report

The authors of the manuscript entitled " Association between selected dietary habits and ipid profile of patients with cardiovascualr disease" aim to investigate the associaition between certain dietary factors and lipid profile in men with diagnosed CVD. The manuscript is easy to read  but the main limitaiton is the completely lack of originality of the research questions. Additionally it was conducted in a relatively small sample size of hospitalized men only. The selection of the study design as the study base is not clear. Results are presenting not clearly mixing previous studies and literature with results. Comments to the results is poor presented.

Abstract:

line 15. please specify the type of questionnaire

line 28: "Our results support the hypothesis that diet is one of the most important 28 determinants in cardiovascular disease and its risk factors". I would suggest to tone down this sentence and stick with what you are testing…..an association 

line 30: "The presented research is very important in the prevention of cardiovascular diseases". Indeed it is but I do not see the originality of the research question.  

INTRODUCTION

line 63-65: "It is well accepted that the consumption of foods rich in saturated fats and cholesterol, like meat, egg yolk and high-fat  dairy products, is associated with increased CVD risk". I would be more careful about this statement. The relationship between these items and CVD has been actually debated and in many studies the association was null.

METHODS

line 79: please specify or add a reference for the "snowball method"

line 82: please specify the type of questionnaire

RESULTS

lines 115-118: the sentences is not fitting with the results section.

lines 122-142 and 144-145: please see the comment above. This is a comment to the results and no a your own result.

line 147: it is not clear for me. How can it be that diffes by sex when you recruited only men. Please clarify better the study participants in the method.

DISCUSSION

I do not see comments in relation to your fonding but just reporting what the other studies found without any comparition also with yuour results.

Author Response

Thank you for all comments to the manuscript and the opportunity to improve them. I think I was able to correct all of the mistakes and take into account all of the comments. Below are the answers of the reviewer’s comments. The changes are marked in the manuscript with the tracking function. Manuscript has been professionally proofread – I attached the certificate.

Answer:

The research questions were designed so that the answers were relevant to the topic of the research.  At the Nitra Cardio-centre are hospitalized patients from a relatively large area in southern Slovakia (6343 m2), but not all hospitalized patients met the conditions of the study and agreed to be included in the research. The condition of the respondents after overcoming the myocardial infarction was also respected too.The methodology was supplemented and modified during the review procedure according to the reviewers' comments, as well as comments on the result.

Abstract:

  1. line 15. please specify the type of questionnaire - questions in closed ended format

Answer:  All of the reviewer's suggestions have been reviewed and corrected in the manuscript. Line 15 was „The data necessary for the detection of dietary habits were obtained by a questionnaire method.“ The sentence has been corrected and completed. The current wording is: „The data necessary for the detection of dietary habits was obtained by a questionnaire method in closed-ended format.”

  1. line 28: "Our results support the hypothesis that diet is one of the most important determinants in cardiovascular disease and its risk factors". I would suggest to tone down this sentence and stick with what you are testing…..an association 

Answer:  All of the reviewer's suggestions have been reviewed and corrected in the manuscript. Line 28 was „Our results support the hypothesis that diet is one of the most important determinants in cardiovascular disease and its risk factors.“ The sentence has been corrected and completed. The current wording is: „Our results support that monitoring the lipid profile is an important determinant in the prevention and treatment of cardiovascular disease.“

  1. line 30: "The presented research is very important in the prevention of cardiovascular diseases". Indeed it is but I do not see the originality of the research question.  

Answer:  All of the reviewer's suggestions have been reviewed and corrected in the manuscript. Line 30 was “The conducted research emphasizes the importance of diet dependence on the improvement of the quality of treatment and nutrition of people with this type of disease.“ The sentence has been corrected and completed. The current wording is: „The conducted research emphasises the importance of diet dependence on the improvement of the quality of treatment and nutrition of people with this type of disease.“

INTRODUCTION

  1. line 63-65: "It is well accepted that the consumption of foods rich in saturated fats and cholesterol, like meat, egg yolk and high-fat  dairy products, is associated with increased CVD risk". I would be more careful about this statement. The relationship between these items and CVD has been actually debated and in many studies the association was null.

Answer:  All of the reviewer's suggestions have been reviewed and corrected in the manuscript. Lines 63 - 65 was „It is well accepted that the consumption of foods rich in saturated fats and cholesterol, like meat, egg yolk and high-fat dairy products, is associated with increased CVD risk.“ The sentence has been corrected and completed. The current wording is: „On the other hand, most recent meta-analyses of randomised trials and observational studies found no beneficial effects of reducing saturated fatty acid (SFA) intake on CVD and total mortality, and instead found protective effects against stroke.“

METHODS

  1. line 79: please specify or add a reference for the "snowball method"

Answer: The selection of the sample was random - there were selected women who were in the hospital in the delivery room and agreed to participate in the study. The incorrect term snowball was probably created during multiple corrections. All of the reviewer's suggestions have been reviewed and corrected in the manuscript. Line 79 was „Patients were selected using the snowball method.“ The sentence has been corrected and completed. The current wording is: Patients were selected using the method of random selection.“

  1. line 82: please specify the type of questionnaire

Answer: All of the reviewer's suggestions have been reviewed and corrected in the manuscript. Line 82 was „The data necessary for the detection of dietary habits were obtained by a questionnaire method. The questionnaire was applied individually by a single interviewer.“ The sentence has been corrected and completed. The current wording is: „The data necessary for the detection of dietary habits was obtained by a questionnaire method in closed-ended format. The questionnaire was applied individually by a single interviewer. The questionnaire contained two parts.“

RESULTS

  1. lines 115-118: the sentences is not fitting with the results section

Answer: After analyzing the snippet under review, however, the authors tend to agree that the lines match the results section. All of the reviewer's suggestions have been reviewed and corrected in the manuscript. Lines 115-118 were „In the prevention of obesity and disorders of lipid metabolism, attention is paid to the regular consumption of meals, with appropriate time intervals between them. This statement is supported by the fact that people with low quantity of food doses per day had increased levels of T-C, and a higher prevalence of obesity and CVD, which was also confirmed by our results.“ The sentences has been corrected and completed. Currently, the sentences are: „In the prevention of obesity and lipid metabolism disorders, attention is paid to the regular consumption of meals, with appropriate time intervals between them. This statement is supported by the fact that people who consume low-quantity food doses per day had increased levels of T-C, and a higher prevalence of obesity and CVD, which was also confirmed by our results.“

  1. lines 122-142 and 144-145: please see the comment above. This is a comment to the results and no a your own result.

Answer: After analyzing the manuscript, the authors are not entirely sure if we properly understood the reviewer's suggestion. In this sentences are our results and discussion. Lines 122-142 and 144-145 were „Previous studies have shown that although a higher daily eating frequency is associated with an increased intake of energy, which comes mainly from carbohydrates and not from fats or proteins. Such studies also found a positive relationship between a higher daily eating frequency and the consumption of nutrients such as dietary fiber, vitamin C, folic acid, calcium, magnesium and iron. Although consumption of red meat is valuable as a source of protein, iron, vitamin B12 and other B vitamins in the human diet, its excessive consumption, as seen by many Western nations, may be associated with obesity, CVD, some types of oncological diseases, diabetes and other non-communicable diseases. The link between meat consumption and the incidence of chronic diseases and mortality has been evaluated in hundreds of observational epidemiological studies over the last few decades. Despite this substantial data, it is not yet clear whether the increased intakes of certain groups of meat (e.g., total, unprocessed or processed red meat) or certain species (e.g., beef or pork) contribute independently to the risk of the disease. We found that pork is the most popular, with 70.88% respondents reported to consume it 1–2 times/week. Escriba-Perez et al.  monitored the consumption of meat in 800 respondents aged 25–75 years.” The sentences has been corrected and completed. Currently, the sentences are: This statement is supported by the fact that people who consume low-quantity food doses per day had increased levels of T-C, and a higher prevalence of obesity and CVD, which was also confirmed by our results. It turned out that most of the respondents (69.25%) consumed 3–4 meals a day (Table 3). Only 25.3% of participants reported 5–6 meals a day. We detected significant differences (p Ë‚ 0.05) between the influence of the number of daily meals on T-C and LDL-C, which were higher in men who consumed 1–2 meals compared with 3–4 or 5–6 meals (Table 3). Previous studies have shown that a higher daily-eating frequency is associated with an increased intake of energy, which comes mainly from carbohydrates and not from fats or proteins. Such studies also found a positive relationship between a higher daily-eating frequency and the consumption of nutrients such as dietary fibre, vitamin C, folic acid, calcium, magnesium and iron.”

  1. line 147: it is not clear for me. How can it be that diffes by sex when you recruited only men. Please clarify better the study participants in the method

Answer:  All of the reviewer's suggestions have been reviewed and corrected in the manuscript. Line 147 was „Red meat consumption frequency also differed between the sexes.“ The sentences has been corrected and completed. Currently, the sentences are: „The most common was chicken, which was consumed by up to 90.87% of respondents at least once a week. The second most frequently consumed was beef (63.62%), and third was pork (52.62%). Escriba-Perez et al. also state that red meat consumption frequency also differed between the sexes.“

DISCUSSION

  1. I do not see comments in relation to your fonding but just reporting what the other studies found without any comparition also with your results.

Answer:  The reviewer's remark was included in the quoted excerpt. The discussion was corrected as suggested by the reviewer.

We would like to thank the Reviewer for his constructive comments. Thanks to them, the article became more valuable, contributing much more to the issues raised in the presented results. Supplementing the publication with the Reviewer's suggestions certainly enriched it. After the review, it was supplemented with several tables, thanks to which the substantive value of the publication was raised. Once again, thank you very much for your right and very substantive comments. Kindly consider the manuscript for publication in your journal.

Reviewer 2 Report

This is a descriptive study showing lipid profile and dietary habits in a total of 800 patients of myocardial infarction or angina pectoris from a Hospital in Slovakia.  The topic is very interesting and the manuscript is well-written; however, I have major suggestions, which are specified below:

  1. The first part of the questionnaire included questions concerning the sociodemographic situation of the subjects, body height and weight, physical activity, use of tobacco and medical history. These are important factors that should be included in the analysis.
  2. The research team used the snowball method to recruit the participants. More details must be provided. What was the targeted number of participants? What was the response rate? Why only men were included, is it a men's hospital? Also please discuss how representative the studied sample was.
  3. This is a wide range of age (20 to 101 years). Please consider providing results stratified by age group.
  4. Please, include in tables 2-7 that values are mean and SD.
  5. In my opinion, many statements in the Results section should be in the Discussion section (e.g. lines 115-118, 123-126, 135-142, etc.).
  6. Some limitations should be mentioned in the manuscript (e.g. the type of analysis, that demonstrates associations but not causation, only men in the sample, etc.).

Author Response

Thank you for all comments to the manuscript and the opportunity to improve them. I think I was able to correct all of the mistakes and take into account all of the comments. Below are the answers of the reviewer’s comments. The changes are marked in the manuscript with the tracking function. Manuscript has been professionally proofread – I attached the certificate.

  1.  The first part of the questionnaire included questions concerning the sociodemographic situation of the subjects, body height and weight, physical activity, use of tobacco and medical history. These are important factors that should be included in the analysis.

Answer:  The reviewer's remark was included in the quoted excerpt. Supplemented in Table 1 and added data in the new Table 2.

  1. The research team used the snowball method to recruit the participants. More details must be provided

Answer: The selection of the sample was random - there were selected women who were in the hospital in the delivery room and agreed to participate in the study. The incorrect term snowball was probably created during multiple corrections.

  1. What was the targeted number of participants? What was the response rate? Why only men were included, is it a men's hospital? Also please discuss how representative the studied sample was

Answer:  The reviewer's remark was included in the quoted excerpt. Data were obtained over four years by the random selection of patients, in the male department of the Nitra Cardo Centre.

  1. This is a wide range of age (20 to 101 years). Please consider providing results stratified by age group.

Answer:  The reviewer's remark was included in the quoted excerpt.

  1. Please, include in tables 2-7 that values are mean and SD.

Answer:  The reviewer's remark was included in the quoted excerpt.

6. In my opinion, many statements in the Results section should be in the Discussion section (e.g. lines 115-118, 123-126, 135-142, etc.).

Answer:  The reviewer's remark was included in the quoted excerpt. Citations in the results section, we consider it a suitable addition and explanation of the importance of the monitored parameters in the given issue. It is similar with other articles in the journal IJERPH

7. Some limitations should be mentioned in the manuscript (e.g. the type of analysis, that demonstrates associations but not causation, only men in the sample, etc.).

Answer: The study is characterized by certain limitations, such as: the type of analysis used, the selection of only men to participate in the study. It would be valuable to examine the issues raised also in the group of women. The article has been enriched with a fragment that has some limitations of publication:

Although this study provided new insight into the relationship between selected eating habits and the lipid profile of patients with cardiovascular disease, it has some limitations. First, the study was conducted only on the population of one country, so it provides detailed information only about that particular population and should be reproduced in other countries. Moreover, the study only looked at men, while in fact the relationship between eating habits is of course also relevant among women. In subsequent studies, it is worth conducting similar studies among people of both sexes. Additionally, comparisons of the results obtained by gender would probably be very valuable. Finally, the study only assessed the declared behaviors that had not been verified by direct observational research techniques.

We would like to thank the Reviewer for his constructive comments.  Thanks to them, the article became more valuable, contributing much more to the issues raised in the presented results. Supplementing the publication with the Reviewer's suggestions certainly enriched it. After the review, it was supplemented with several tables, thanks to which the substantive value of the publication was raised. Once again, thank you very much for your right and very substantive comments. Kindly consider the manuscript for publication in your journal.

Reviewer 3 Report

Revision of manuscript titled: „Association between selected dietary habits and lipid profile of patients with cardiovascular disease” by KopeÄŤková et al.

The revised manuscript aimed to assess the association between some dietary habits and lipid profile of patients diagnosed with cardiovascular diseases.

The manuscript is interesting and provides valuable contribution in the research field of human nutrition but at a present form cannot be published and needs a major revision.

Specific comments to the Authors:

- the manuscript needs a careful language check!

INTRODUCTION:

- lines 42-44: the sentence “Mainly they include high serum total cholesterol (TC) and low-density lipoprotein cholesterol (LDL-C), high-lipoprotein cholesterol (HDL-C)” is incorrect. TC, LDL-C and HDL-C refers to serum concentration. There is no a term “high-lipoprotein cholesterol”, but “high-density lipoprotein cholesterol”;

- the name of hospital, where the patients were diagnosed should be unified (the authors indicated Nitra Cardio Centre, Kardiocentrum Nitra or Cardiocenter Nitra for the same hospital).

MATERIALS AND METHODS:

- please clearly describe the inclusion criteria for the survey participants;

- How were monitored the body weight and height of responders? Please provide the relevant methodology with references;

- lines 82-84 and 90-92: the sentences are repeated!

- line 93: the name of University is written in capital letters!

- the information regarding the test of normality of data distribution must be completed in the text. Were the all analyzed parameters characterized by normal distribution?

RESULTS:

- I have serious doubts regarding the description of the results, because were presented improperly. Due to the huge range of participants’ age (20 -101 years), the results should be analyzed taking account the age subgroups!

- Based on the BMI value, please provide the data of number (or percentage) of responders characterized by normal, lower and excessive body weight;

- Section 3 (Results) is, in fact, a combination of the description of the results and discussion and, at the same time, the next section of the manuscript is discussion. These sections must be re-written (the Results separately from discussion or Results and Discussion together);

- Table 4: please explain, why the answers regarding the frequency of meat consumption were limited only to three options that do not fully relate to accurate situation?

- Tables 3-7: The first column of the tables must include the number and percentage of respondents!

- Why the question about egg consumption was limited only to 3 answers per week? What about the responders who declared the consumption less than once a week? Why did you give an answer “0-5 eggs/week” and did not indicate “no consumption” as separate option?

- There were no option the answer “no consumption” regarding the frequency of milk and diary products consumption?

- I strongly suggest performing more accurate statistical analysis, apart from analysis of variance (e.g. multivariable regression adjusted to different factors, like BMI, tobaco use, age etc.;

- DISCUSSION and CONCLUSION must be matched to the corrected description of the results!

Author Response

Thank you for all comments to the manuscript and the opportunity to improve them. I think I was able to correct all of the mistakes and take into account all of the comments. Below are the answers of the reviewer’s comments. The changes are marked in the manuscript with the tracking function. Manuscript has been professionally proofread – I attached the certificate.

- the manuscript needs a careful language check!

Answer: They were deleted by mistake. The article was subjected to language correction in an external company. A translation, proofreading and editorial certificate is included.

INTRODUCTION:

  1. lines 42-44: the sentence “Mainly they include high serum total cholesterol (TC) and low-density lipoprotein cholesterol (LDL-C), high-lipoprotein cholesterol (HDL-C)” is incorrect. TC, LDL-C and HDL-C refers to serum concentration. There is no a term “high-lipoprotein cholesterol”, but “high-density lipoprotein cholesterol”;

Answer: All of the reviewer's suggestions have been reviewed and corrected in the manuscript. Linees 42 -44 was „Mainly they include high serum total cholesterol (TC) and low-density lipoprotein cholesterol (LDL-C), high-lipoprotein cholesterol (HDL-C), as well as hypertension, smoking, diabetes, obesity and sedentary way of life.“ The sentence has been corrected and completed. The current wording is: „Mainly, they include: high serum total cholesterol (TC), low-density lipoprotein cholesterol (LDL-C), high-density lipoprotein cholesterol (HDL-C), as well as hypertension, smoking, diabetes, obesity and sedentary lifestyle.“

  1. the name of hospital, where the patients were diagnosed should be unified (the authors indicated Nitra Cardio Centre, Kardiocentrum Nitra or Cardiocenter Nitra for the same hospital).

Answer:  The reviewer's remark was included in the quoted excerpt

MATERIALS AND METHODS:

  1. please clearly describe the inclusion criteria for the survey participants;

Answer:  The reviewer's remark was included in the quoted excerpt

  1. How were monitored the body weight and height of responders? Please provide the relevant methodology with references;

Answer:  The reviewer's remark was included in the quoted excerpt

  1. lines 82-84 and 90-92: the sentences are repeated!

Answer: All of the reviewer's suggestions have been reviewed and corrected in the manuscript. Lines 82 - 84 was „The data necessary for the detection of dietary habits were obtained by a questionnaire method. The questionnaire was applied individually by a single interviewer. The questionnaire was anonymous.“ The sentence has been corrected and completed. The current wording is: „.The data necessary for the detection of dietary habits was obtained by a questionnaire method in closed-ended format. The questionnaire was applied individually by a single interviewer. The questionnaire contained two parts. The first part included questions concerning the socio-demographic situation of the subjects, physical activity, use of tobacco and medical history. The second part of the questionnaire concerned the analysis of selected dietary habits, including the number of the consumed meals, their regularity, snacking between meals, types of snacks consumed and the eating frequency of selected groups of food products. The questionnaire was anonymous; its completion was voluntary with only one response to be circled for each question. The questionnaire was compiled by the Department of Human Nutrition (Slovak University of Agriculture in Nitra) and approved by the Nitra Cardio Centre.“

  1. line 93: the name of University is written in capital letters!

Answer:  The reviewer's remark was included in the quoted excerpt

  1. 107: - the information regarding the test of normality of data distribution must be completed in the text. Were the all analyzed parameters characterized by normal distribution?

Answer:  All of the reviewer's suggestions have been reviewed and corrected in the manuscript. Line 107 was “The data was checked for normality. Data were expressed as mean ± standard deviation (SD). Statistical comparisons between groups were undertaken using one-way analysis of variance (one-way ANOVA), followed by Tukey's post hoc test. Significance was accepted when p < 0.05. The program Statistica Cz 10 (TIBCO Software, Inc., Palo Alto, CA, USA), and MS Excel 2007 (Microsoft Corporation, Redmond, WA, USA) were used.” The sentence has been corrected and completed. The current wording is: Statistical comparisons between groups were undertaken using one-way analysis of variance (ANOVA), followed by Tukey's post hoc test. Significance was accepted when p < 0.05. The programs Statistica Cz 10 (TIBCO Software, Inc., Palo Alto, CA, USA) and MS Excel 2007 (Microsoft Corporation, Redmond, WA, USA) were used.

RESULTS:

  1. I have serious doubts regarding the description of the results, because were presented improperly. Due to the huge range of participants’ age (20 -101 years), the results should be analyzed taking account the age subgroups!

Answer:  The reviewer's remark was included in the quoted excerpt

  1. Based on the BMI value, please provide the data of number (or percentage) of responders characterized by normal, lower and excessive body weight

Answer:  The reviewer's remark was included in the quoted excerpt

  1. Section 3 (Results) is, in fact, a combination of the description of the results and discussion and, at the same time, the next section of the manuscript is discussion. These sections must be re-written (the Results separately from discussion or Results and Discussion together);

Answer: Citations in the results section, we consider it a suitable addition and explanation of the importance of the monitored parameters in the given issue. It is similar with other articles in the journal IJERPH

  1. Table 4: please explain, why the answers regarding the frequency of meat consumption were limited only to three options that do not fully relate to accurate situation?

Answer:  The reviewer's remark was included in the quoted excerpt. Based on the revision of the results, the data (no consumption) were supplemented

  1. Tables 3-7: The first column of the tables must include the number and percentage of respondents!

Answer:  The reviewer's remark was included in the quoted excerpt

  1. Why the question about egg consumption was limited only to 3 answers per week? What about the responders who declared the consumption less than once a week? Why did you give an answer “0-5 eggs/week” and did not indicate “no consumption” as separate option?

Answer:  The reviewer's remark was included in the quoted excerpt. Based on the revision of the results, the data (no consumption) were supplemented

  1. There were no option the answer “no consumption” regarding the frequency of milk and dairy products consumption?

Answer:  The reviewer's remark was included in the quoted excerpt. Based on the revision of the results, the data (no consumption) were supplemented

  1. - I strongly suggest performing more accurate statistical analysis, apart from analysis of variance (e.g. multivariable regression adjusted to different factors, like BMI, tobaco use, age etc.;

Answer:  The pulication was supplemented with statistical analyzes. The obtained results are presented in the tables (10-12) added after the Reviewer's suggestion. The authors thank you very much for the valuable suggestion that enriched the manuscript.

  1. DISCUSSION and CONCLUSION must be matched to the corrected description of the results!

Answer:  The reviewer's remark was included in the quoted excerpt

We would like to thank the Reviewer for his constructive comments.  Thanks to them, the article became more valuable, contributing much more to the issues raised in the presented results. Supplementing the publication with the Reviewer's suggestions certainly enriched it. After the review, it was supplemented with several tables, thanks to which the substantive value of the publication was raised. Once again, thank you very much for your right and very substantive comments. Kindly consider the manuscript for publication in your journal. Kindly consider the manuscript for publication in your journal.

Round 2

Reviewer 1 Report

The manuscript investigating the relationship between specific dietary factors and lipids is improved. However, the authors are still not presenting the results section in a proper scientific way. Several part of the results section refer to previous studies. I would recommend to revise it.

Author Response

We would like to thank the Reviewer for his constructive comments. Thanks to them, the article became more valuable, contributing much more to the issues raised in the presented results. Supplementing the publication with the Reviewer's suggestions certainly enriched it. Chapters: Results and Discussion have been corrected and reorganized according to the Reviewer's signatures. The authors hope that this time the Reviewer's comments have been satisfactorily addressed.

Reviewer 2 Report

I think that the manuscript has not been significantly improved. Important factors as age group, body height and weight, physical activity, use of tobacco or medical history are slightly included in the analysis and those new results are not added to the discussion. The study has some important weakness in the design and the execution. Besides the manuscript, the study itself have several problems. The sample doesn’t follow stronger recruitment criteria and the results are not in line with the discussion. Regarding the recruitment methods, the author’s response is that “there were selected women who were in the hospital in the delivery room and agreed to participate in the study”. That doesn’t make any sense to me. I also asked other questions as the targeted number of participants or the response rate and I didn’t get an answer. I still think that many statements in the Results section should be in the Discussion section. In my opinion, these are some aspects that makes this manuscript no acceptable for publication.

Author Response

Thank you for all comments to the manuscript and the opportunity to improve them. I think I was able to correct all of the mistakes and take into account all of the comments. Below are the answers of the reviewer’s comments. The changes are marked in the manuscript with the tracking function. Manuscript has been professionally proofread – I attached the certificate.

Reviewer 2

  1. Regarding the recruitment methods, the author’s response is that “there were selected women who were in the hospital in the delivery room and agreed to participate in the study”.

The authors would like to thank the Reviewer for detailed, yet critical comments. In answering Recent, of course, in the preparation of the answer to the above objection, there was a misunderstanding and incorrect translation (the sample was selected at random - women who were in the hospital in the delivery room were selected and agreed to participate in the study. Probably, during many corrections, the incorrect term "snowball" was created ".). For which the authors apologize very much.

  1. What was the targeted number of participants? What was the response rate? Why only men were included, is it a men's hospital? Also please discuss how representative the studied sample was

Answer:  The answer that should be given to the Reviewer is of course the following: 958 men took part in the study. During the data review process, we found that some respondents' responses were incomplete and had to be excluded from the survey. The response rate was 83.5%. And this is the only correct and unequivocal response to the Reviewer's suggestion and instructions. We apologize once again for the previous wording.

  1. Some limitations should be mentioned in the manuscript (e.g. the type of analysis, that demonstrates associations but not causation, only men in the sample, etc.).

Answer: The study is characterized by certain limitations, such as: the type of analysis used, the selection of only men to participate in the study. Other limitations of the publication include not including all issues related to socio-demographic factors in the analyzes, which may result in the picture of the situation not being fully complete. The above-mentioned limits, which characterize this work, will certainly be included in subsequent publications, which are prepared by the team. It would be valuable to examine the issues raised also in the group of women.

The article has been enriched with a fragment that has some limitations of publication:

Although this study provided new insight into the relationship between selected eating habits and the lipid profile of patients with cardiovascular disease, it has some limitations. First, the study was conducted only on the population of one country, so it provides detailed information only about that particular population and should be reproduced in other countries. Moreover, the study only looked at men, while in fact the relationship between eating habits is of course also relevant among women. In subsequent studies, it is worth conducting similar studies among people of both sexes. Additionally, comparisons of the results obtained by gender would probably be very valuable. Finally, the study only assessed the declared behaviors that had not been verified by direct observational research techniques.

The Results and Discussion chapters have been revised and properly redrafted.

We would like to thank the Reviewer for his constructive comments.  Thanks to them, the article became more valuable, contributing much more to the issues raised in the presented results. Supplementing the publication with the Reviewer's suggestions certainly enriched it. After the review, it was supplemented with several tables, thanks to which the substantive value of the publication was raised. Once again, thank you very much for your right and very substantive comments. Kindly consider the manuscript for publication in your journal.

Reviewer 3 Report

Reviewer decision: minor revision

The Authors have improved the manuscript according to my suggestions, but some issues still need to be corrected.

According to me, the Authors did not address properly to my suggestion:

”Section 3 (Results) is, in fact, a combination of the description of the results and discussion and, at the same time, the next section of the manuscript is discussion. These sections must be re-written (the Results separately from discussion or Results and Discussion together)”

In the RESULTS section, the authors should describe the results obtained in the presented study. There is no need to refer to the results of other authors in this section, because this is a part of Discussion section.

Table 12: I suggest to place the superscript asterisks to the correlation coefficients, if P<0.05.

Author Response

We would like to kindly inform you that all elements that could disturb the unambiguity of the chapters Results and Discussion have been properly ordered and re-edited.

We absolutely agree with the comments of the Reviewer 2. Therefore, the statement related to the reference to the own results obtained was extracted from the discussion. The Discussion chapter has been improved and its purpose is to discuss the obtained results with other authors presenting similar research.

The authors would like to thank the Reviewer for the detailed analysis of the publication, it certainly contributed to the improvement of the quality of the text and its substantive value.

The authors express their hope that the corrections made are consistent with and that they satisfy the reviewer's expectations.

Table 12 has also been corrected in line with the Reviewer's suggestions.
